# Implementation of an Internet of Things Architecture to Monitor Indoor Air Quality: A Case Study During Sleep Periods

**DOI:** 10.3390/s25061683

**Published:** 2025-03-08

**Authors:** Afonso Mota, Carlos Serôdio, Ana Briga-Sá, Antonio Valente

**Affiliations:** 1Department of Engineering, School of Sciences and Technology, Universidade de Trás-os-Montes e Alto Douro (UTAD), 5000-801 Vila Real, Portugal; afonsomagmota@hotmail.com (A.M.); cserodio@utad.pt (C.S.); anas@utad.pt (A.B.-S.); 2Center ALGORITMI, Universidade do Minho, Campus de Azurém, 4800-058 Guimarães, Portugal; 3CQ-VR (Center of Chemistry of Vila Real), Universidade de Trás-os-Montes e Alto Douro, 5000-801 Vila Real, Portugal; 4CRIIS (Centre for Robotics in Industry and Intelligent Systems), INESC-TEC—Institute for Systems and Computer Engineering, Technology and Science, 4200-465 Porto, Portugal

**Keywords:** indoor air quality, ESP32, MQTT, InfluxDB, CO_2_, ubiquitous, IoT, IAQ monitoring

## Abstract

Most human time is spent indoors, and due to the pandemic, monitoring indoor air quality (IAQ) has become more crucial. In this study, an IoT (Internet of Things) architecture is implemented to monitor IAQ parameters, including CO_2_ and particulate matter (PM). An ESP32-C6-based device is developed to measure sensor data and send them, using the MQTT protocol, to a remote InfluxDBv2 database instance, where the data are stored and visualized. The Python 3.11 scripting programming language is used to automate Flux queries to the database, allowing a more in-depth data interpretation. The implemented system allows to analyze two measured scenarios during sleep: one with the door slightly open and one with the door closed. Results indicate that sleeping with the door slightly open causes CO_2_ levels to ascend slowly and maintain lower concentrations compared to sleeping with the door closed, where CO_2_ levels ascend faster and the maximum recommended values are exceeded. This demonstrates the benefits of ventilation in maintaining IAQ. The developed system can be used for sensing in different environments, such as schools or offices, so an IAQ assessment can be made. Based on the generated data, predictive models can be designed to support decisions on intelligent natural ventilation systems, achieving an optimized, efficient, and ubiquitous solution to moderate the IAQ.

## 1. Introduction

As quality of life continues to improve, and considering the pandemic context [1], the importance of breathing environments has gained increasing attention from researchers in the twenty-first century [2] due to its impact on health and well-being. Due to modern lifestyles, approximately 80–90% of human life is spent indoors, implying that individuals are more exposed to indoor pollutants than outdoor pollutants [3,4,5,6]. It is essential to study the types of pollutants that degrade indoor air quality, as well as their sources so that their presence can be mitigated, thereby creating a satisfactory indoor environment and enhancing the building occupants’ well-being [6].

This deterioration of indoor air quality (IAQ) has recently been addressed as IAP (Indoor Air Pollution), and it can become worse if the necessary ventilation for air renewal is not guaranteed. Several indoor air contaminants, including Volatile Organic Compounds (VOCs), ozone (O_3_), carbon monoxide (CO), carbon dioxide (CO_2_), and particulate matter (PM), have been shown to affect IAQ adversely [7]. These substances can be found either in outdoor or indoor environments. For instance, carbon monoxide (CO) can be produced when cooking or heating—from gas heaters and stoves, fireplaces, and furnaces—and can exceed recommended values within the household, which can have adverse health effects, such as cardiovascular and neurobehavioral issues at low concentrations and unconsciousness or death at high concentrations [8].

Carbon dioxide (CO_2_) presence indoors can be generated through combustion reactions such as cooking, heating, and smoking but also from human metabolism. The most impactful source in offices and transportation vehicles is metabolic CO_2_ release due to respiration. Studies that exposed subjects to different concentrations of this gas were performed, showing that individuals exposed to CO_2_ levels of 945–1400 ppm (parts per million) had a decrease in cognitive function and decision-making ability, and exposure to 2000–3000 ppm may cause health symptoms such as headaches or tiredness. These symptoms correlate with hypercapnia, which is when the brain receives CO_2_-rich blood and its function is affected [1,3,9]. Therefore, it is crucial to continuously monitor and manage indoor CO_2_ levels to prevent health issues and to safeguard the cognitive function of the building occupants.

Particulate matter, on the other hand, can have major negative effects on the heart and lungs if inhaled [8]. The particles are a mixture of solid and liquid droplets within the environment that cannot be seen by the naked eye [10] and can be classified based on the diameter: ${\mathrm{PM}}_{10}$—coarse particles (<10 microns); ${\mathrm{PM}}_{2.5}$—fine particles (<2.5 microns); and ${\mathrm{PM}}_{0.1}$—ultrafine particles (<0.1 microns). Fine particles (${\mathrm{PM}}_{2.5}$) are currently the focus of most respiratory health research, but even smaller particles, particularly ultrafine particles (${\mathrm{PM}}_{0.1}$), may be more toxic, as they can penetrate cell membranes [11,12]. The indoor generation of these particles is associated with human activity, like the burning of materials, which is present in everyday activities such as cooking and heating but also from cleaning products. Outdoors, road traffic is one of the prominent factors in developed countries [13].

With this, to effectively understand the state of indoor air quality, a device with the capability to measure the approached pollutants is essential [6]. Additionally, integrating this device with an IoT (Internet of Things) system is crucial for continuous data collection to achieve the real-time and remote monitoring of the studied environment. Therefore, this work contributes to an end-to-end measurement and data analysis infrastructure:A device with appropriate sensors for IAQ measurement and wireless connectivity to send data;Usage of open-source database instances to effectively store data;Tools to query the database to allow further data analysis, which are used to investigate bedroom CO_2_ levels during sleep, using the scenario of a slightly open door and another of a closed door.

## 2. Literature Review

For the last few years, and with the pandemic surge, indoor air quality has become a major focus in research communities, namely, real-time monitoring devices that enable remote data assessments to be performed. Various studies have explored different sensors and IoT communication protocols for IAQ monitoring.

In 2018, Benammar et al. [14] developed a Zigbee WSN (Wireless Sensor Network) for IAQ monitoring, comprising several electrochemical devices to measure SO_2_ (sulfur dioxide), NO_2_ (nitrogen dioxide), O_3_ (ozone), CO (carbon monoxide), Cl_2_ (chlorine), an NDIR (Non-Dispersive Infrared) sensor to capture CO_2_ concentrations (INE20-CO2P-NCVSP [15]), and a BME280 [16] to retrieve temperature and humidity. These sensors are connected to an ATMega1281 [17] and a XBee PRO radio to enable low-power wireless communication. To agglomerate data from end devices, a Zigbee gateway using a Raspberry Pi 4 [18] is implemented, where a Python script is used to control the flow of Zigbee messages and relay data to the Internet (using HTTP (HyperText Transfer Protocol), with an Emoncms API (Application Programmable Interface), which is a web server with the purpose of logging data).

In 2019, Hapsari et al. [19] pointed out that indoor air quality can be more dangerous than that of outdoor air. Therefore, using low-cost sensors, a system that measures temperature, humidity, CO_2_, and dust levels was developed. To monitor data in real-time, an IoT architecture was presented, where the end device sends data using Wi-Fi and MQTT (Message Queuing Telemetry Transport) to a MySQL database. The end device was developed using ESP8266 [20], a particulate matter sensor DSM501a [21], a temperature and humidity sensor DHT11, and an electrochemical sensor MQ-135 [22], which can detect different gas concentrations (NH_3_—ammonia, NO_x_—nitrogen oxides, CO_2_, among others) based on the resistance value obtained. Typically, this conversion and post-processing is computationally heavy and requires a lot of calibration, which might not be ideal for low-powered end devices that run processing constraint applications. Findings show that values from IAQ are normal, specifically in larger rooms. However, the presented work still defends that poor IAQ harms people and that stricter regulations should be implemented.

Reinforcing that air quality impacts health and comfort, and in an urge to monitor air quality issues and implement corrective actions in buildings, Jose et al. [23] developed an IoT sensor to monitor IAQ. The sensors comprised are BME680 [24], a metal-oxide sensor developed by Bosch that is used widely nowadays, SPG30 [25], and CCS811 [26]. These sensors can detect different VOCs and the equivalent CO_2_—a composite measure that represents the concentration of CO_2_ based on detecting other gases—however, measurements may not be accurate [27,28]. Data aggregated in the end device are transmitted using LoRa because it is a suitable long-distance low-power communication technology. However, data storage and visualization are not approached in this research. It was also found that VOCs and CO_2_eq (CO_2_ equivalent) values were high and made calls for social actions, indicating that people should know the risks of poor IAQ, look for signs, and maintain ventilated spaces if needed.

In 2020, the works presented in Kanal et al. [29,30] implemented a BLE (Bluetooth Low Energy) distributed sensor network of Silicon Thunderboard Sense 2 devices [31]. To analyze specific air quality parameters such as tVOC (total Volatile Organic Compounds) and CO_2_eq, this board features a CCS811 sensor [26]. Values from this sensing unit are sent using BLE to a central gateway that will redirect traffic to the Internet. This central device is an SBC (Single-Board Computer) running PyGatt, a module written in Python that allows reading from and writing to BLE devices that implement the Generic Attribute Profile (GATT). In this case, each end device acts as a GATT server, and the gateway interacts with the servers to retrieve sensor data. Afterward, data are sent using the SNMP (Simple Network Management Tool) protocol to Cacti , a network monitoring and management tool, where data can be visualized.

To minimize the power consumption of end devices that monitor the environment, Agbulu et al. [32] developed a ULP-IoT (Ultra-low-power IoT System) for IAQ monitoring that provides readings of CO_2_, fine particles (${\mathrm{PM}}_{2.5}$), carbon monoxide, humidity, and temperature. This sensing unit incorporates low-cost sensors MICS-5524 [33] (CO) and MQ-135 [22] (CO_2_ and ${\mathrm{PM}}_{2.5}$) connected to an ESP32 [34] with deep sleep enabled that sends data via HTTP to a central server in a Raspberry Pi 3 [18] when in active mode. However, this type of electrochemical sensor (MQ-135 [22]) has a calibration process of 24 h that consumes a fair amount of current to heat its resistance, requires it to be in clean air, and is relatively complex [35,36]. Liu et al. [37] designed an IoT air quality device and continuously tested in residential buildings for a continuous month in winter. The sensor developed is Zigbee-enabled, connected to a gateway, which transmits data to a cloud server using GPRS (General Packet Radio Service) or 4G. In this case, CO_2_ is measured using an NDIR sensor to achieve high resolution, particulate matter is captured by PMS5003 [38], and data are sent using a Zigbee wireless module (CC2630 chip [39]). All of these modules are managed via UART (Universal Asynchronous Receiver/Transmitter) using a microcontroller unit (STM32 [40]). With these modules, measurements are taken in rooms with varying occupancy and door states (open/closed), highlighting the crucial role of ventilation in maintaining indoor air quality, namely during sleep.

Serroni et al. [41] designed “Comfort Eye”, an IoT air quality sensor to address the intrusiveness and expensiveness of IEQ monitoring. This device uses an NDIR sensor to measure CO_2_—SCD30 [42]—and SPS30 [43] to measure (${\mathrm{PM}}_{2.5}$). These are connected to a PyCOM W01 board [44]. Data are transmitted to a remote server using MQTT and then are delivered to a MySQL database. In a pre- and post-renovation study in a Polish nursery, KPIs (Key Performance Indicators) indicated higher CO_2_ levels before renovation and better PM levels. However, after the renovation, there were slight improvements in IAQ due to the implementation of an indicator LED in the board, which provided real-time feedback by prompting occupants to open the windows to improve ventilation. This emphasizes how straightforward communication with an IAQ monitor can enhance air quality by influencing the occupant’s behavior.

Bristol Research and Innovation Laboratory (BRIL) developed an IoT system to monitor indoor environmental conditions at Cardiff University using a LoRaWAN (Long-Range Wireless Area Network) topology [45]. To measure the environment, there are I2C sensors such as SCD41 (NDIR CO_2_ sensor [46]) and BME680 (a Bosch environmental metal-oxide sensor [16]) but also a PMS5003 [38] via UART. A gateway agglomerates data from these described end devices (recurring to an LPWAN—Low-Power Wireless Area Network concentrator) and forwards them to a ChirpStack LoRaWAN Network Server and database, which is local, in Cardiff University, and can be accessed via VPN for remote monitoring.

Concentrations of CO_2_ are correlated with the presence of aerosols within a room and can be used as an indicator for airborne transmission of SARS-CoV-2 [47], and many IAQ monitoring tools have been developed. However, they are multi-parameter (some of which are not needed), tend to consume high amounts of energy, are large and fairly expensive, ending up being unpractical. Therefore, Kuncoro et al. [47] developed a low-power, low-cost, and compact end device that enables CO_2_ monitoring using an NDIR sensor (SCD30 [42]) via I2C (Inter-Integrated Circuit). A dashboard where data can be observed was elaborated. Sensor performance was evaluated, demonstrating that it is reliable with minimal error and that these devices can help reduce virus transmission.

In the same pandemic context, Yasin et al. [48] stated that the latest public health crisis increased interest in indoor air quality monitoring. In the practical case, off-the-shelf devices were used and instilled in an IoT architecture—Netatmo indoor and outdoor modules [49] were deployed (and other devices) and connected to their specific IoT cloud platform. Then, a middleware solution using Node-RED was utilized to communicate with each device cloud platform (using its specific API) and retrieve sensor values to store these in an InfluxDB database. This is an intelligent solution because multiple devices from different manufacturers use various technologies, and a single middleware can seamlessly integrate them all.

The year 2023 has shown major improvements relative to the works shown in the previous years. Barros et al. [50] tested an air quality sensor at the University Fernando Pessoa and deployed it in Alpendorada High School (Northern Portugal) revealing that high CO_2_ concentrations frequently exceed recommended levels during class, which indicates poor ventilation. However, the authors pointed out that outdoor air influences indoor air quality; therefore, high PM outdoor levels might require protective measures such as closing windows. This device was developed with ESP32 [34] running ESP-HOME, a platform that simplifies configuring these microcontrollers through YAML (Yet Another Markup Language) files. This platform allows communication with IAQ sensors via UART: MH-Z19 NDIR CO_2_ sensor [51] and PMS5003 PM sensor [38]. The IoT architecture is based on the Home Assistant (HA) platform and add-ons, such as InfluxDB for storing data and Grafana for providing visualization. Although HA is more focused in home automation, the SchoolAIR project takes a slightly different approach to scale the architecture. For this, the project employs edge nodes as air quality sensors, fog nodes as HA instances with local applications (Raspberry Pi boards [18]), and cloud node which is a central HA instance that aggregates all data and shows them on dashboards. Using the same technologies to store and visualize measurements, the author [52] equipped a Raspberry Pi board with Node-RED, InfluxDB, and Grafana software. This setup allowed to subscribe, archive, and display MQTT messages from end devices. Their hardware solution for the air quality modules was the SCD30 CO_2_ sensor [42] and BME680 IAQ sensor [24]. Furthermore, they could forecast CO_2_ steady-state concentration levels with minimal error recurring to an LSTM (Long Short-Term Memory) prediction model.

Air quality sensors can also be employed to monitor pig farming facilities. Arulmozhi et al. [53] developed a low-cost system with a Raspberry Pi 4 board with several sensors, including PMS5003 [38] and MH-Z19 [51] as previously referred but also an Enviro+ shield [54] that contains additional environmental parameters. Sensor modules store data locally but also send measurements to a MySQL server by establishing an HTTP connection and communicating via REST API in JSON format. The system was tested in two barns with pigs, and reliability checks were performed by distribution analysis and statistical evaluation, which revealed measurements were similar in their distributions, validating the sensor performance. Taamté et al. [55] employed a similar CO_2_ sensor together with Zigbee wireless modules, conducted thorough calibration procedures, and executed comparative studies that validated these sensor’s performance.

All-in-one sensor solutions are also available, which Chou et al. [4] utilized to monitor a surgical Intensive Care Unit (ICU) under visitation restrictions during the recent pandemic. An IoT device with a PTQS1005 sensor [56] (includes CO_2_, ${\mathrm{PM}}_{2.5}$, and other environmental measurements) and ESP8266 [20] sends data to a MySQL server and then the data can be visualized in a Grafana instance. Findings suggest that the number of visitors and employees on the job has an impact on CO_2_ levels in the ICU (Intensive Care Unit). The developed system can enable health professionals to monitor levels of air pollution.

In sum, the focus on IAQ devices has intensified, particularly due to the pandemic, which has led to the development of IoT-compatible real-time monitoring systems. Different types of wireless communications have been utilized, particularly Wi-Fi, LoRa, and Zigbee, which demonstrate the adaptability of modern IAQ monitors. For precise CO_2_ measurements, NDIR sensors have been utilized more recently [14,37,41,45,47,50], and are reliable and have high accuracy. Some authors use relational database to store data, while others use time-series databases designed to efficiently manage large volumes of time-stamped data [57].

This research article aims to develop a device with state-of-the-art sensors to monitor indoor air quality using highly optimized, professional-grade, and scalable firmware development to achieve complete control and customization. The state-of-the-art components, such as the Sensirion SCD40 CO_2_ sensor and the Panasonic SN-GCJA5 particulate matter sensor, are widely recognized for their accuracy and reliability. Our key innovations include the integration of these sensors with the ESP32-C6 microcontroller and the development of a scalable and flexible IoT architecture using MQTT protocol and InfluxDB for real-time data storage and visualization. Afterward, data can be consulted and queried using Python programming language [58]. This study focuses on demonstrating the initial capabilities of the developed IoT system to monitor IAQ parameters. In this work, to support the system evaluation, preliminary observations on the impact of different ventilation modes are presented. However, it is expected that devices will become able to collect environmental data to aid in poor IAQ mitigation strategies.

## 3. Materials and Methods

The architecture of the IoT system developed in this work was designed to be capable of monitoring air quality in a home, office, or school environment, and to transmit data to remote computers. To do so, an end device based on an ESP32 microcontroller, chosen for its versatility and Wi-Fi capabilities, publishes messages of SCD40 sensor temperature, humidity, CO_2_, and SN-GCJA5 sensor particulate matter (${\mathrm{PM}}_{0.1}$, ${\mathrm{PM}}_{2.5}$ and ${\mathrm{PM}}_{10}$) to a remote MQTT broker. Remotely on another machine, an InfluxDB instance is running, and the Telegraf add-on subscribes to air quality monitors (end device) messages and stores them in the database, where they can be visualized. A diagram of the end-to-end proposed architecture and utilized technologies can be observed in Figure 1. In this work, state-of-the-art components such as the sensors, controlling unit, and database instances are configured accordingly and put together in an IoT infrastructure to measure IAQ. The presented key innovations include the integration of these modules to create a scalable IoT architecture using the MQTT protocol for real-time data exchange and InfluxDB for efficient data storage and visualization. The implementation of custom firmware using the ESP-IDF (Espressif IoT Development Framework) enables complete control and end-device optimization, providing an innovative solution for wireless continuous IAQ monitoring.

### 3.1. End Device

The end device is responsible for collecting IAQ data and transmitting them to a remote database for storage. It should be wirelessly compatible with different types of environments, such as offices and universities, to send data.

#### 3.1.1. Hardware

The end device is built around the ESP32-C6 microcontroller [59] to achieve the requirements, specifically the development kit board. This unit, manufactured by Espressif Systems (Shanghai) Co., Ltd., Shanghai, China, features a RISC-V processor and supports SPI (Serial Peripheral Interface), UART, I2C, and I2S, among others, which are some of the most used protocols to communicate with off-the-shelf sensors. It has the 802.15.4 radio, meaning it is Thread and Zigbee compatible, but also 802.11b/g/n/ax radio, being compatible with Wi-Fi 6. The Target Wake Time (TWT) feature of the 802.11ax standard enables building battery-connected devices based on ESP32-C6 to last for years. This device also accelerates matter device development, a standardized open-source home automation protocol [60].

In this edge node, the SCD40 CO_2_ sensor from Sensirion [61] is utilized—an NDIR sensor that provides accurate CO_2_ measurements by emitting infrared light and then detecting the absorption of a specific wavelength on a gas sample. With compact size, low power consumption, and built-in temperature and humidity compensation, this solution is suitable for space constraints and low-cost devices. Providing precise CO_2_ data, this sensor can be useful in different scenarios, such as minimizing virus transmission and occupancy estimation [62]. The SN-GCJA5 sensor from Panasonic [63] is used to detect particulate matter, which can detect particles of different sizes (${\mathrm{PM}}_{0.1}$, ${\mathrm{PM}}_{2.5}$, and ${\mathrm{PM}}_{10}$). The sensor sends laser beams that are intersected with particles in the air and then detects the created angle of scattered light using photodiodes. These fed to an MCU inside the sensor, which runs an optimized algorithm to interpret these scattering patterns and convert them into a mass-density value μg/$m^{3}$. To achieve wired communication with both sensors, SCD40 and SN-GCJA5, a I2C bus is implemented, where ESP32-C6 is the master, and sensors are slaves [63]. The complete requirements for the end device can be seen in Figure 2.

#### 3.1.2. Firmware

This ESP32-C6 based air quality monitoring end device is developed using ESP-IDF (Espressif IoT Development Framework). This SDK (Software Development Kit) is made available by the chip manufacturer Espressif Systems and provides a robust set of tools and libraries that enable efficient application development. With this, I2C custom drivers are written for both sensors to achieve communication with ESP32-C6 hardware interfaces. FreeRTOS is utilized for multitasking, with a specific task dedicated to reading sensor values regularly and publishing them to an MQTT broker, using the MQTT client and Wi-Fi libraries provided by ESP-IDF. By leveraging this framework’s peripheral management libraries, reliable data transmission can be achieved. All environment settings and installation procedures are taken according to the Espressif guides [64]. A main application is written based on firmware modules described in the next subsections. Then, it is built and flashed to the device.

Before initializing the sensors, a set of functions is implemented to handle I2C communication. The initialization function, sensor_init, sets up the configuration of the I2C master interface with data and clock I/O pins (SDA and SCL) and installs the I2C driver as a master device. This function ensures that the I2C bus is correctly established and ready for communication. To interact with SCD40, the scd40_read_measurement function is designed as shown in Algorithm 1. It initializes an I2C command link that builds up a series of operations: it starts by configuring the microcontroller as an I2C master, then initiates communication with the sensor by sending its I2C address (0x62) and the read command (0xEC05), followed by a stop command (lines 4 and 5). A short delay ensures that the sensor has enough time to process (line 6). Then measurements are read with the same process—a command link is created to perform the read operation, retrieving 9 bytes from the sensor that contains raw data (lines 7–12). The integrity of these measurements is ensured by computing a Cyclic Redundancy Check (CRC) for each data segment in the firmware and comparing these CRC values to the ones received in the data packet from the sensor (lines 14 and 15). Subsequently, raw sensor data are converted into meaningful values using the manufacturer’s indications (lines 17 to 19). Furthermore, sensor data are calibrated by enabling Automatic Self-Calibration (ASC) function from SCD40 sensor (0x2416 command), ensuring accuracy by continuously adjusting the baseline carbon dioxide concentrations over time. With this feature, the lowest measured concentration over seven days (which is typically around 400 ppm) is assumed as the representation of outdoor carbon dioxide, compensating for the sensor drifts and maintaining its accuracy over time.
**Algorithm 1:** SCD40 sensor measurement reading
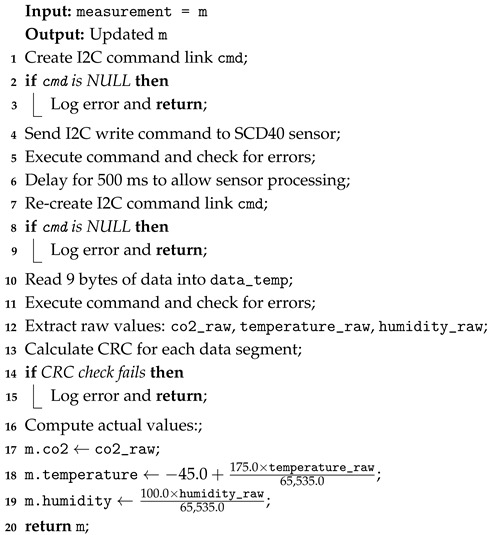


To communicate with the SN-GCJA5 sensor, each parameter (${\mathrm{PM}}_{0.1}$, ${\mathrm{PM}}_{2.5}$ and ${\mathrm{PM}}_{10}$) mass-density value contains 4 bytes. Therefore, four sequential registers must be read from the sensor to each. Algorithm 2 describes the implemented process, and I2C functions are implied. Therefore, to obtain a 4-byte sequence, a function called read_32bit_value_from_register is created using read register operations, and its input parameter is the initial register (described in lines 1 to 7). Then, specific functions like read_pm1, read_pm2_5 and read_pm10 use the 32-bit reading function to extract and convert to a floating-point mass-density particulate matter value (lines 8 to 19). This approach enables precise data acquisition and conversion from the sensor’s raw output. This raw output is calibrated with the SN-GCJA5 auto-calibration function that runs automatically. The sensor monitors the laser diode’s light output and the fan’s rotational speed over time, and the buit-in MCU adjusts the components to maintain the performance within the specified bounds. When the bounds are exceeded, this MCU provides an optimal correction value and starts software corrections automatically to improve the performance and sensor lifetime.
**Algorithm 2:** Sensor initialization and data reading    **Input**: None    **Output**: pm1_value, pm2_5_value, pm10_value  1**Read 32-bit Value from Register:**  2  Read byte 0 from start_reg_addr;  3  Read byte 1 from start_reg_addr + 1;  4  Read byte 2 from start_reg_addr + 2;  5  Read byte 3 from start_reg_addr + 3;  6  Combine the 4 bytes into a 32-bit value:;  7    ∗value ← (byte3<< 24) | (byte2<< 16) | (byte1<< 8) | byte0;  8**Read ${\mathrm{PM}}_{0.1}$ Value:**  9  Call Read 32-bit Value from Register with ${\mathrm{PM}}_{0.1}$ register address;10  Convert the raw 32-bit value to a float by dividing by 1000;11  Store the result in pm1_value;12**Read ${\mathrm{PM}}_{2.5}$ Value:**13  Call Read 32-bit Value from Register with ${\mathrm{PM}}_{2.5}$ register address;14  Convert the raw 32-bit value to a float by dividing by 1000;15  Store the result in pm2_5_value;16**Read ${\mathrm{PM}}_{10}$ Value:**17  Call Read 32-bit Value from Register with ${\mathrm{PM}}_{10}$ register address;18  Convert the raw 32-bit value to a float by dividing by 1000;19  Store the result in pm10_value;

The approach used to establish Wi-Fi and MQTT connections is detailed in Algorithm 3. A function called wifi_init is used to initialize the Wi-Fi, which sets up components like NVS (Non-Volatile Storage), TCP/IP (Transmission Control Protocol/Internet Protocol), and event loops. In here, the device is configured to initialize with the station mode (STA) and connect to a predefined WiFi network (lines 11 to 28). A function called event_handler processes the WiFi events—it reconnects automatically if needed and sets an event bit if an IP address is obtained, so the program can wait for a network connection before proceeding (lines 1 to 10). Subsequently, to initiate a MQTT connection, a mqtt_init function is implemented (lines 36–40). Here, credentials are configured, and the MQTT client is initiated with an event handler, particularly to log the events (connection and disconnection to broker—lines 29–35). Finally, a mqtt_publish function is created to publish MQTT packets, with the payload and topic specified as input parameters (lines 41–43). The utilized MQTT broker is not disclosed in this article, but to recreate this experience, an open source would work accordingly.
**Algorithm 3:** Wi-Fi and MQTT initialization and communication
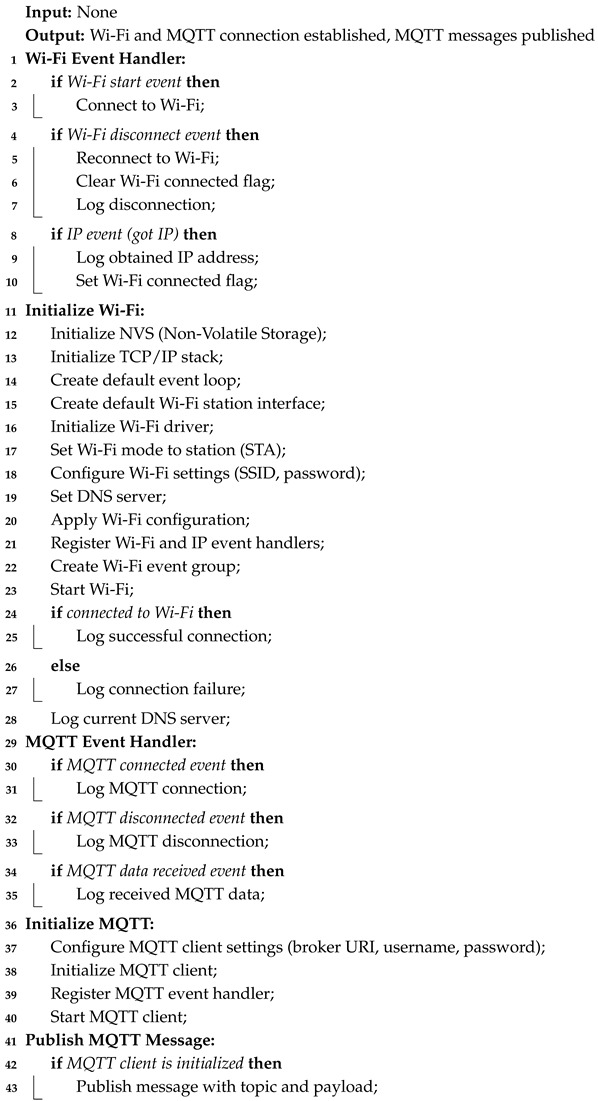


### 3.2. Data Storage Using InfluxDB and Telegraf

InfluxDB is an open-source time series database that handles large volumes of time-stamped data. It is well suited for IoT applications requiring continuous data collection from multiple end-nodes. In this application, InfluxDBv2 is utilized, which has an integrated UI (User Interface) for configuration and monitoring data. This instance is configured according to the manufacturer’s guidelines and runs as a service on a remotely located Windows machine. For data aggregation from IoT sensors, a plugin-driven server agent built for collecting and reporting metrics, Telegraf, is used as an intermediary between the MQTT broker and database instance. A configuration file is required to provide input plugins for subscribing data from a specific MQTT broker URL, port and topic, along with other possible parameters as seen in Listing 1. This plugin collects sensor data and delivers them to a bucket in InfluxDB, configured in the “outputs” section of the file [65].

**Listing 1.** Telegraf configuration file.1[[outputs.influxdb_v2]]2 urls = [“:<INFLUXDB_URL>:<INFLUXDB_PORT>”]3 token = “:<INFLUXDB_TOKEN>:”4 organization = “:<INFLUXDB_ORG>:”5 bucket = “:<INFLUXDB_BUCKET>:”6
7[[inputs.mqtt_consumer]]8 alias = “consumer_json_sensordata”9 servers = [“:<BROKER_URL>:<BROKER_PORT>”]10 topics = [“:<MQTT_TOPIC>:”]11 username = “:<MQTT_USERNAME>:”12 password = “:<MQTT_PASSWORD>:”13 data_format = “json”


InfluxDBv2 has advanced features, such as an integrated data explorer that allows users to run Flux query language to flexibly obtain and transform data from the database. Furthermore, it provides simple dashboarding tools that allow the real-time monitoring of entities without having to elaborate complex graphical interfaces. This setup supplies a robust and scalable solution to store, query, and analyze the IoT device’s data.

### 3.3. Querying Database for Data Analysis Using Python

Once data are stored in the database, they can be queried for different periods using Python. This is useful for inspecting the data, like perceiving trends, running statistical analysis, or generating reports. To set up this environment in the same machine running InfluxDB, Python 3.11 is installed together with a required library influxdb-client. A developed Python script (Algorithm 4) demonstrates how connections can be made to an InfluxDB 2.x instance, then queries are run to fetch specific data, and results are visualized with Matplotlib version 3.9.1. To achieve this, the script defines connection parameters for the InfluxDB instance, which includes URL, token, organization, and bucket name where data are stored. This ensures that this application can authenticate and communicate correctly with the database instance. This is demonstrated in lines 1 to 9.

After connecting to InfluxDB using Python, the Flux query language specifies how data are to be retrieved from a bucket. To achieve this, the written query filters data by a specific topic (sensor/data) and field (CO_2_, temperature, humidity, PMx, etc.). Furthermore, the start and stop variables are manually configured, which defines the time range of the data query. Data are aggregated into 30 min intervals using the aggregateWindow function, obtaining a mean value of the measurements for each interval. This allows the query to run faster for visualization, but raw values can be obtained if necessary (for example, for statistical analysis). A query can be observed in Listing 2, and is sufficiently flexible to accept different bucket names and start/stop dates using F strings. After running the Flux query, the returned data are processed by being converted into a Pandas data frame to enable further manipulation as implied in lines 16–18 of Algorithm 4. Finally, Matplotlib is utilized to plot each parameter level over the specified period, providing clear visuals of how measurements fluctuate over time as described in lines 19–27.

**Listing 2.** Flux query used in Python script.1from(bucket: “{bucket}”)2      |> range(start: {start}, stop: {stop})3      |> filter(fn: (r) => r[“_field”] == “CO2”)4      |> aggregateWindow(every: 30m, fn: mean, createEmpty: false)5      |> yield(name: “mean”)
**Algorithm 4:** Fetch and plot CO_2_ data from InfluxDB    **Input**: None    **Output**: Plots for CO_2_ Levels on Day 1 and Day 2 with Door Status  1**Initialize InfluxDB Connection:**  2  Set url to InfluxDB server URL;  3  Set token to InfluxDB authentication token;  4  Set org to organization name;  5  Set bucket to bucket name;  6  Create InfluxDB client using url, token, and org;  7  Initialize query API from the client;  8**Define Time Ranges:**  9  Set queries with time ranges for Day 1 (September 3) and Day 2 (September 4);10**Fetch Data Function:**11  **Input:**start, stop12  Construct query to fetch CO_2_ data from bucket within the time range;13  Execute the query using query API;14  Collect data from query results;15  Convert collected data to DataFrame;16**Fetch Data for Both Days:**17  Call fetch function for Day 1 and store in df_day_1;18  Call fetch function for Day 2 and store in df_day_2;19**Plot Data:**20  Create a figure with two subplots;21  **Plot for Day 1 (September 3, Door Closed):**22    Plot CO_2_ levels from df_day_1 on the first subplot;23    Label the plot to indicate “Door Closed”;24  **Plot for Day 2 (September 4, Door Open):**25    Plot CO_2_ levels from df_day_2 on the second subplot;26    Label the plot to indicate “Door Open”;27  Adjust layout and display the plot;


This methodology to obtain values from the database using Flux query language is particularly useful given that it can help developers, data analysts, and other stakeholders to identify patterns that might provide insights about indoor air quality in specific time periods. For instance, occupation and ventilation behaviors can be analyzed, and data-driven decision-making can be performed to, for example, increase natural ventilation levels and/or optimize energy consumption, enhancing building performance. Users can benefit from this toolkit to manage and analyze time-series data stored in InfluxDB instances. However, this is just a starting point, which could evolve into more sophisticated data analysis, such as forecasting indoor air quality by predicting each pollutant pattern.

### 3.4. Environment Setup and Monitored Scenarios

The developed air quality monitoring is deployed in a university student’s room, located in Vila Real, Portugal. The room has approximate dimensions of 5.5 m in length, 4 m in width, and  3 m in height, creating a total volume of 66 $m^{3}$. The sensor is mounted approximately 1.5 m above the ground, on the eastern wall. The window is located on the southern side of the room, while the door is located on the northern side.

Two scenarios are considered to test this IoT architecture and gather insights about the indoor air quality of this room, specifically during night time:Door closed: The door is fully closed on the first night to create a sealed environment with the minimal air exchange possible.Door slightly open: The door is left slightly open, around 10 cm, allowing air exchange between the room and the hallway.

These implemented setups allowed for the observation of how door positioning influences IAQ, especially CO_2_ levels.

## 4. Results and Discussion

As described in Section 3, a ESP32-C6 microcontroller is equipped with two sensors: SN-GCJA5 and SCD40. The sensors are connected to the microcontroller I2C bus and supplied with electrical power (3.3 V for SCD40 and 5 V to SN-GCJA5). The ESP32-C6 development kit board has 3.3 V and 5 V output and is connected to a conventional USB power source (5 V). The complete setup can be observed in Figure 3.

To do so, the board is connected to the local Wi-Fi network to enable continuous data transmission. MQTT is utilized to send data in real time to a remote machine containing the database. All the credentials (and additional information, such as the MQTT topic) are hard coded onto the device for demonstration purposes, which is insecure.

In the log present in Listing 3, the behavior of the ESP32-C6 air quality monitoring device can be observed. After the normal initial bootup process, which is not represented here to reduce log size, the device starts the Wi-Fi module as the STA (station) mode and undergoes the process of connecting to an AP (Access Point) (lines 1 and 2). Detailed logs are also omitted to reduce the verbosity of the presented log file. After a few seconds, the device obtained an IP address (192.168.1.202) from the local DHCP server, with a specific network mask and gateway (lines 3–7). A simple function to verify network connectivity is implemented, which ensures that the device could communicate over the Internet (line 8). Then, the MQTT event handler registers the MQTT Connected event, meaning the device has successfully authenticated and communicates with the MQTT Broker (line 9). Following that, the sensor task begins, and measurements from the SCD40 and SN-GCJA5 sensors take place, followed by a published MQTT message containing sensor values (lines 11–17). These data are structured into a JSON string using dynamic memory allocation techniques in the firmware before being published to the MQTT broker.
**Listing 3.** Device logs after boot process.1WIFI_MQTT: Wi-Fi initialization completed2[…] - Wi-Fi-specific behavior until AP connection3esp_netif_handlers: sta ip: 192.168.1.202, mask: 255.255.255.0, gw:
   192.168.1.2544WIFI_MQTT: Got IP: 192.168.1.2025WIFI_MQTT: Connected to Wi-Fi network6WIFI_MQTT: Using DNS Server: 192.168.1.2547wifi:<ba-add>idx:1, ifx:0, tid:0, TAHI:0x10050de, TALO:0x713e571c, (ssn:1,
   win:64, cur_ssn:1), CONF:0xc00000058WIFI_MQTT: Network connectivity test succeeded9WIFI_MQTT: MQTT Connected10main_task: Returned from app_main()11SCD40: CO2: 992 ppm12SCD40: Temperature: 28.71 °C13SCD40: Humidity: 43.85%14PM_SENSOR: PM0.1: 12.617 µg/m^3^15PM_SENSOR: PM2.5: 14.190 µg/m^3^16PM_SENSOR: PM10: 15.963 µg/m^3^17MQTT: Published message with msg_id=50401


### 4.1. Data Storage in InfluxDBv2 and Trends

This section details the storage, visualization, and analysis of environmental data using InfluxDB v2. Utilizing its dashboard capabilities, developers can monitor key indoor air quality parameters such as CO_2_, temperature, humidity, and particulate matter in real time. The data, collected over a week and stored via Telegraf, highlight trends like the correlation between CO_2_ levels and human presence, temperature and humidity fluctuations, and particulate matter variations linked to human activity. This supports the validation of the proposed IoT infrastructure, as well as the device sensor readings.

#### 4.1.1. Data Visualization

As previously mentioned, InfluxDB v2 features dashboard functionalities, which allow developers to monitor and analyze data directly in the database without the need for additional software for data visualization. In Figure 4, the environmental parameters measured in seven days by the end device can be observed—CO_2_ (ppm), temperature (°C), relative humidity (%), and particulate matter (μg/$m^{3}$ for ${\mathrm{PM}}_{0.1}$, ${\mathrm{PM}}_{2.5}$, and ${\mathrm{PM}}_{10}$). These data are stored in a bucket using Telegraf—the input plugin connects to the MQTT broker, subscribes to this data-specific topic, and delivers it to the bucket. Each sensor reading is stored as a point in the bucket with the timestamp as the key. In the present dashboard, the concept of downsampling to display data is utilized, where data are aggregated at less granular intervals (in this case, half-hour mean values from raw measurements). This process accelerates queries over extensive time periods by minimizing the volume of data that needs to be processed during the query [65].

#### 4.1.2. Overall Trends

Carbon dioxide concentrations are correlated with human presence. In this case, high CO_2_ concentrations are consistent with nighttime periods, having higher peaks when the sleep period is higher and passing the most common 1000 ppm CO_2_ threshold for brief periods [3,66].

Temperature and humidity present an inversely proportional relation as expected. This means that relative humidity tends to come down when the temperature goes up, and vice versa. This is because when the air temperature increases, the air can hold more moisture, meaning relative humidity will decrease, and conversely [67].

All particulate matter parameters (${\mathrm{PM}}_{0.1}$, ${\mathrm{PM}}_{2.5}$, and ${\mathrm{PM}}_{10}$) have similar distributions, although no visual correlation can be performed with other parameters. However, these values do tend to rise when there is human activity but not directly during sleep, probably only due to particle resuspension with movement [13]. The usage of deodorant also elevates these values. Nonetheless, 24-h exposure levels of 15 μg/$m^{3}$ for ${\mathrm{PM}}_{2.5}$ and 45 μg/$m^{3}$ for ${\mathrm{PM}}_{10}$ specified in AQG (Air Quality Guidelines) from WHO (World Health Organization) are not exceeded [68].

### 4.2. Data Analysis with Python

In this use case, the objective is to compare indoor CO_2_ levels on two different days during nighttime, focusing on understanding the impact of the door state (slightly open/closed) during sleep. The proposed hypothesis is that this IoT device can detect that keeping the door slightly open will lead to lower CO_2_ levels, contrary to a closed door, which results in higher levels due to limited airflow.

To analyze CO_2_ daily trends, a Python script fetches and processes data from the InfluxDBv2 database. This enables us to visualize and compare CO_2_ levels for different periods [58,65], which are submitted to different door state conditions. As described in Section 3, this script starts by establishing a connection to the database using the influxdb-client library, allowing querying using Flux language to retrieve data from the database. Two similar queries are written, differing only in the time range.

The obtained measurements can be seen in Figure 5. With the door closed, CO_2_ concentrations rise linearly with two different slopes. In this specific case, the lack of ventilation leads to a continuous accumulation of CO_2_ in the room, achieving values above 1000 ppm for 3 h. Comparatively, with the door slightly open, the overall rise is slower, with lower peak concentrations, and does not exceed the recommended values. This suggests that minimal indoor ventilation, and not necessarily with the outdoors directly, is adequate for moderating CO_2_ accumulation. Therefore, the recommended CO_2_ levels are not exceeded, indicating that the slightly open-door scenario provides a healthier environment, especially during sleep.

Sleeping with an opened door, however, might pose privacy or security challenges [69]. People may find the trade-off between better IAQ and the referred challenges irrelevant, keeping doors and windows closed. As this gas cannot be seen or felt, the individuals of today’s society will probably not even address this issue [70]. To tackle these problems, it is necessary to explore autonomous ventilation solutions, especially intelligent natural ventilation systems, so our indoor environments can ubiquitously refresh the air when it is required [5,71]. Occupancy levels can significantly influence the indoor air quality (IAQ), and in poorly ventilated spaces, the number of occupants may be inferred by observing the steady-state behavior of CO_2_ levels [52,72]. Furthermore, the occupancy patterns might be regular, especially in schools and offices, with similar peak and off-peak times. With this valuable information, predictive models can be designed to support the evolution of ventilation systems, moving towards more intelligent and ubiquitous solutions. These sensing devices should also be compatible with the most well-known smart home systems to reduce individuals’ barriers to adoption [60].

By leveraging an IoT architecture comprising an ESP32-C6-based device, MQTT protocol, and InfluxDB with Telegraf, it is demonstrated that indoor environmental parameters can be measured, stored, and visualized in real-time. Using the Python scripting language to query the database allows a more in-depth analysis of measurements, offering insights relative to IAQ. Although this developed work focuses on gathering data of two night time period for comparison, the ability to query the database using software allows us to build faster and more autonomous statistical analysis and predictive modeling. By including this set of tools, future applications could allow us to better understand and forecast the air humans breathe indoors.

## 5. Conclusions

This study successfully demonstrated a real-time indoor air quality monitoring system with remote storage, visualization, and analytics. The implemented end device is based on ESP32-C6, and its manufacturers’ SDKs were employed for firmware development. Drivers for sensors were developed to retrieve CO_2_, temperature, humidity, and particulate matter data, and drivers for wireless and application layer communications. By using MQTT protocol, Telegraf and InfluxDB, and Python, data obtained from the device can be saved, visualized, and analyzed. This set of tools paves the way for further development in the IAQ examination.

Using the referred system, the findings highlight the importance of bedroom ventilation at night, even if it is not with the exterior air. Measurements of CO_2_ values were analyzed for two different sleeping periods, one with the door closed and another with the door slightly open, and results showed that CO_2_ levels rose steeply and exceeded the recommended levels with the closed door. Overall, sleeping with an open door may improve the air quality but might pose privacy or security issues. While the study was limited to a single occupant and sleeping, the implemented architecture can be deployed in other indoor environments and activities. This is a step closer to developing air quality forecasting models. Integrating these models with ventilation systems can lead to healthier indoor environments for occupants.

In conclusion, the real-time IAQ monitoring system presented in this research has proven to be a valuable tool for managing indoor environments. It has the potential for broader applications, and refinement, especially in firmware. In the future, the objective is to build and deploy more devices in various environments to gain more accurate insights into IAQ and how space is utilized. Furthermore, another pertinent objective is building predictive modeling systems leveraging this IoT architecture.

## Figures and Tables

**Figure 1 sensors-25-01683-f001:**
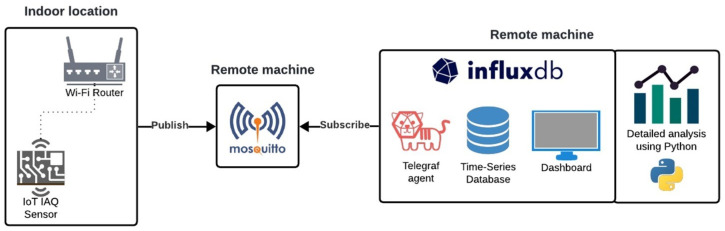
Indoor air quality monitoring IoT architecture.

**Figure 2 sensors-25-01683-f002:**
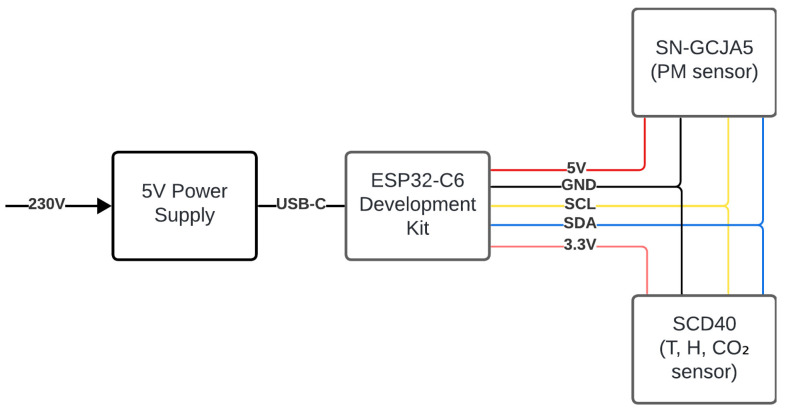
Requirements for the end device.

**Figure 3 sensors-25-01683-f003:**
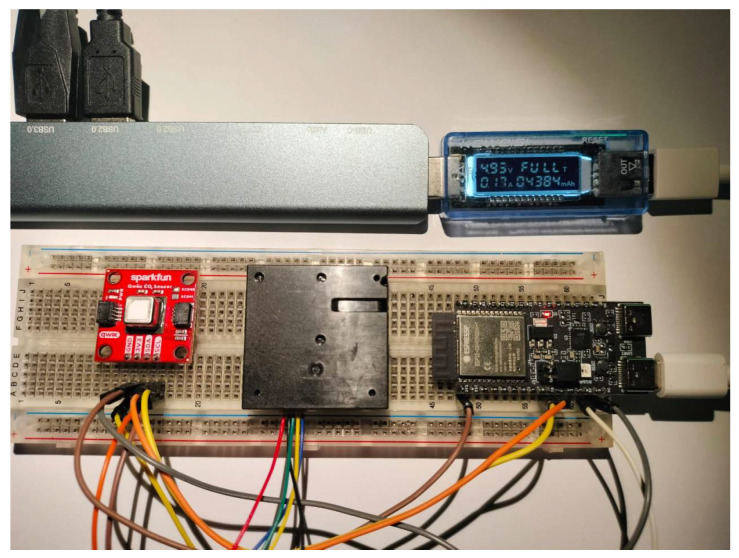
Developed end device.

**Figure 4 sensors-25-01683-f004:**
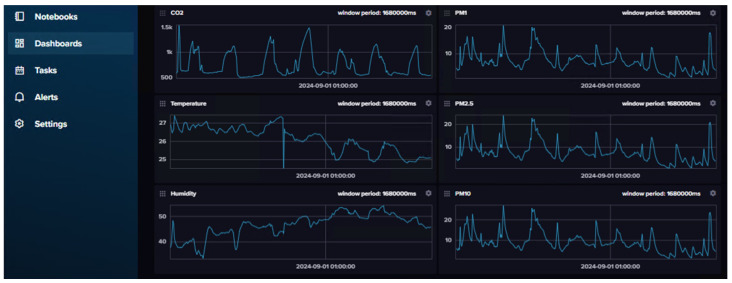
InfluxDB UI with dashboard—end-device sensor data over 7 days (27 August 2024 to 3 September 2024)—CO_2_ (ppm), temperature (°C), relative humidity (%), and particulate matter (μg/$m^{3}$ for ${\mathrm{PM}}_{0.1}$, ${\mathrm{PM}}_{2.5}$, and ${\mathrm{PM}}_{10}$).

**Figure 5 sensors-25-01683-f005:**
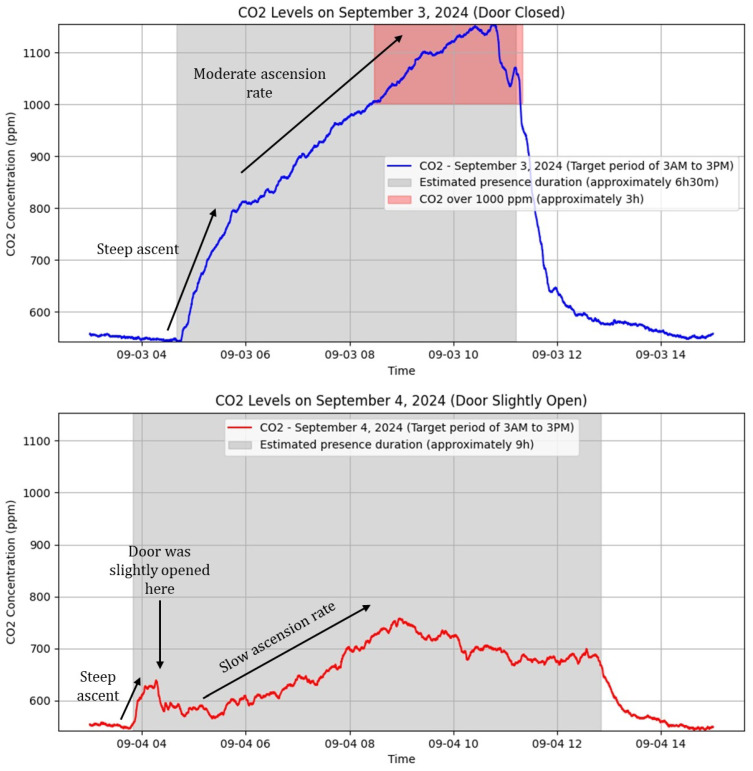
CO_2_ analysis of two days using Python—September 3 with the door closed; September 4 with the door slightly open (10 cm).

## Data Availability

The raw data supporting the conclusions of this article will be made available by the authors on request.

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
