# Peer review of "Implementation of an Internet of Things Architecture to Monitor Indoor Air Quality: A Case Study During Sleep Periods"

_sensors, 2025, doi:10.3390/s25061683_

Round 1
Reviewer 1 Report
Comments and Suggestions for Authors
In this study, an IoT (Internet of Things) architecture is implemented to monitor IAQ parameters, including CO2 and Particulate Matter (PM), which is contributing to assist in regulating indoor air quality. The content of this study is detailed and complete, and has certain innovation. However, the following questions need to be answered by the authors:
1. In this study, carbon dioxide, temperature, humidity and particulate matter were monitored. Which instruments were used for each substance? Please add them to the Materials and Methods.
2. In Figure 4, is it possible to read the concentration of each substance in real time? What is the concentration unit? Is the measured data calibrated?
3. There is too much content in Literature Review. It is recommended to simplify it.
4. Please check for writing errors, for example, CO2 in line 400 and in figure 5. Is there an error in line 87?
Reviewer 2 Report
Comments and Suggestions for Authors
- Trying to identify the objectives of the present study it seems that the primary objective is to implement an IoT architecture to build a monitoring system for indoor CO2 and Particulate Matter (PM) as well as temperature and humidity. This is also the only objective presented explicitly by the authors. This can be perceived by a reader that the system (a) will be described in sufficient detail and (b) will be evaluated for its performance, strength and reliability demonstrated also in the field.
Concerning (a) : there is a extensive description of the system components without making sufficiently clear what is state of the art and what is innovation.
Concerning (b) : It seems that there is no clear focus on the system evaluation. Instead another implicit objective has been introduced : evaluate the environment (i.e. the different ventilation modes) with the help of the system rather than evaluate the system using field data.
- With respect to the presented objective, there is no justification based on the state of the art vs innovation border line . Please expand.
- It might be better to exploit from the field experimental data, only the part that supports the system evaluation. It is suggested the environment evaluation part to be used to support future publication (s) addressing probably the interesting problem of indoor IAQ mitigation strategies by using such advanced systems .
- Based on the above comments all individual chapters need to be modified accordingly. Concerning Chapter 3, it is true that the system is described in detail making the life easier of the ‘constructor’ that has the ambition to build the same system. On the other hand it is obvious that the average reader needs to understand what is state of the art and what is new. Therefore is preferable that the state of the art to be described briefly enough by giving all necessary references and concentrate more on the innovations as well as the proper system evaluation.
Round 2
Reviewer 2 Report
Comments and Suggestions for Authors
The authors have complied to a large extent to the reviewers comments. There are no other substantial comments to be made.